# COVID-19 Clinical Profiles and Fatality Rates in Hospitalized Patients Reveal Case Aggravation and Selective Co-Infection by Limited Gram-Negative Bacteria

**DOI:** 10.3390/ijerph19095270

**Published:** 2022-04-26

**Authors:** Kamaleldin B. Said, Ahmed Alsolami, Safia Moussa, Fayez Alfouzan, Abdelhafiz I. Bashir, Musleh Rashidi, Rana Aborans, Taha E. Taha, Husam Almansour, Mashari Alazmi, Amal Al-Otaibi, Luluh Aljaloud, Basmah Al-Anazi, Ahmed Mohialdin, Ahmed Aljadani

**Affiliations:** 1Department of Pathology, College of Medicine, University of Ha’il, Ha’il 55476, Saudi Arabia; s201806142@uoh.edu.sa (A.A.-O.); s201803639@uoh.edu.sa (L.A.); s201800476@uoh.edu.sa (B.A.-A.); 2Genomics, Bioinformatics and Systems Biology, Carleton University, 1125 Colonel-By Dr, Ottawa, ON K1S 5B6, Canada; 3ASC, McGill University, 21111 Lakeshore Rd, Montreal, QC H9X 3L9, Canada; 4Department of Internal Medicine, College of Medicine, University of Ha’il, Ha’il 55476, Saudi Arabia; a.alsolami@uoh.edu.sa (A.A.); a.aljadani@uoh.edu.sa (A.A.); 5Department of Microbiology, King Salman Specialist Hospital, Ha’il 55476, Saudi Arabia; safiamoussa89@yahoo.com (S.M.); has-lab-kssh@moh.gov.sa (F.A.); 6Department of Physiology, College of Medicine, University of Ha’il, Ha’il 55476, Saudi Arabia; ah.bashir@uoh.edu.sa; 7Ministry of Health, Hail Region, Ha’il 55476, Saudi Arabia; mreshidi@moh.gov.sa; 8Department of Community Medicine, Faculty of Medicine, University of Ha’il, Ha’il 55476, Saudi Arabia; r.alghamdi@uoh.edu.sa; 9Department of Epidemiology, John Hopkins Bloomberg School of Public Health, Baltimore, MD 21205, USA; ttaha1@jhu.edu; 10Health Management Department, College of Public Health and Health Informatics, University of Ha’il, Ha’il 81481, Saudi Arabia; h.almansour@uoh.edu.sa; 11College of Computer Science and Engineering, University of Ha’il, Ha’il 81481, Saudi Arabia; ms.alazmi@uoh.edu.sa; 12Department of Surgery, College of Medicine, University of Ha’il, Ha’il 55476, Saudi Arabia; a.mohialdin@uoh.edu.sa

**Keywords:** SARS-CoV-2 pandemic, empirical-antimicrobial therapy, nosocomial resistance, selective infections, mortality

## Abstract

Bacterial co-infections may aggravate COVID-19 disease, and therefore being cognizant of other pathogens is imperative. We studied the types, frequency, antibiogram, case fatality rates (CFR), and clinical profiles of co-infecting-pathogens in 301 COVID-19 patients. Co-infection was 36% (*n* = 109), while CFR was 31.2% compared to 9.9% in non-co-infected patients (z-value = 3.1). Four bacterial species dominated, namely, multidrug-resistant *Klebsiella pneumoniae* (37%, *n* = 48), extremely drug-resistant *Acinetobacter baumannii* (26%, *n* = 34)*,* multidrug-resistant *Eschericia. coli* (18.6%, *n* = 24), and extremely drug-resistant *Pseudomonas aeruginosa* (8.5%, *n* = 11), in addition to other bacterial species (9.3%, *n* = 12). Increased co-infection of *K. pneumoniae* and *A. baumannii* was associated with increased death rates of 29% (*n* = 14) and 32% (*n* = 11), respectively. *Klebsiella pneumoniae* was equally frequent in respiratory and urinary tract infections (UTI), while *E. coli* mostly caused UTI (67%), and *A. baumannii* and *P. aeruginosa* dominated respiratory infections (38% and 45%, respectively). Co-infections correlated with advance in age: seniors ≥ 50 years (71%), young adults 21–49 years (25.6%), and children 0–20 years (3%). These findings have significant clinical implications in the successful COVID-19 therapies, particularly in geriatric management. Future studies would reveal insights into the potential selective mechanism(s) of Gram-negative bacterial co-infection in COVID-19 patients.

## 1. Introduction

The global community has been witnessing one of the most devastating coronavirus pandemics of all time. The enhanced epidemicity and mutability of Severe Acute Respiratory Syndrome Coronavirus 2 (SARS-CoV-2) have created significant gaps in the mechanisms of pathogenicity, where the last resorts for control relied on supportive therapies. Thus, in the absence of widely available specific therapies, most efforts rely on supportive treatment, biocontainment protocols, and mitigating the role of co-infections that aggravate the disease.

To date, the significance of superinfections or co-infections in aggravating SARS-CoV-2 infection has not been widely studied. The role of co-infections in SARS-1 and Middle East respiratory syndrome (MERS) outbreaks was well-characterized [1,2,3]. For COVID-19 patients, the reported prevalence was variable at the lower rates, reaching to as high as 50% among non-survivors. The common bacterial isolates included *Streptococcus pneumoniae*, *Staphylococcus aureus*, *Klebsiella pneumoniae*, *Mycoplasma pneumoniae*, *Chlamydia pneumonia*, *Legionella pneumophila*, and *Acinetobacter baumannii*. In addition, candida species, *Aspergillus flavus*, as well as viruses such as influenza, coronavirus, rhinovirus/enterovirus, parainfluenza, metapneumovirus, influenza B virus, and human immunodeficiency virus, were found [3].

Many reports indicated the scarcity of SARS-CoV-2 co-infections. Most of these reports were in China [4,5,6,7,8,9,10], and only three were in the United States of America (USA) [3,11,12] and one each in Singapore and Italy (16.7%) [13,14]. The study populations ranged from 18 to 5700 cases, and while none in Singapore had a secondary infection, 50% of non-survivors in China did. Similar findings during the early stages of COVID-19 in Wuhan reported a rate of 16% of secondary infections in hospitalized patients [15], which was higher among the non-survivor group than survivors (50% vs. 1%). The latter two groups, in the aforementioned study, had significant differences in white blood cell counts and absolute values of lymphocytes. In Spain, only 7.2% co-infections (71/989) were reported during hospitalization, inconsistent with the higher prevalence rates in other viral pandemics [16]. In the study, community-acquired co-infection was mainly caused by *S. pneumoniae* and *S. aureus*, whereas hospital-acquired superinfections were mostly caused by *Pseudomonas aeruginosa* and *Escherichia coli*. Patients with co-infections had worse outcomes and the overall mortality was 9.8% (97/989).

Due to the lack of knowledge on co-infection, many decisions regarding antibiotic therapy to COVID-19 patients were made with limited clinical experience and scientific evidence [16]. The need for combination therapy was based on previous assumptions carried over from influenza pandemics which resulted in poor prognosis [2]. The relatively lower levels and variabilities in reported co-infections in SARS-CoV-2 compared to other pandemics made it difficult for a universal consensus on co-infections [17]. For instance, in the United Kingdom (UK), in a total of 836 patients, only 3.2% had an early co-infection, where *S. aureus* was the most common pathogen [18]. Similarly, in France, 28% of bacterial co-infections in severe SARS-CoV-2 pneumonia patients in the intensive care unit (ICU) were mostly *S. aureus*, *Hemophilus influenzae*, *S. pneumoniae*, and Enterobacteriaceae [19]. Although empiric monotherapy was encouraged, confirmation by larger studies to assess the real prevalence and the predictors of co-infection together with its prognostic impact on critically ill patients was imperative [19].

The mechanisms underlying the low bacterial co-infections in COVID-19 compared to other viral pandemics are not understood. It is plausible that enhanced immune reactions and macrophage hyperactivations may have played a role in addition to the empiric antibiotic use and quarantine, which limits exposure. Future studies will be needed to elucidate the role of factors involved in decreasing superinfections. Following the original recommendation made for the treatment of influenza co-infections [20], a decision was made on empirical therapy during COVID-19 in China. In the first 41 cases of SARS-CoV-2-infected patients, they all received empirical antibiotic treatment, while 93% (*n* = 38) received antiviral therapy [4]. This was modified during the second case series of 99 patients, where antibiotic, antiviral, and antifungal agents were administered in 71%, 76%, and 15% of patients, respectively [5]. The third case series comprised of 138 patients requiring intensive care unit admission. Most of the patients received antiviral therapy (oseltamivir, 124 (89.9%)), while many received antibacterial therapy (moxifloxacin, 89 (64.4%), ceftriaxone, 34 (24.6%), and azithromycin, 25 (18.1%)) [21]. Similarly, in a large case series of 1099 patients, most (58.0%) received intravenous antibiotics, and 35.8% received oseltamivir [22]. Nevertheless, according to the recommendations of National Institutes of Health (NIH) and the Infectious Diseases Society of America guidelines, not enough data on co-infections are available to establish a consensus on empiric treatment [23] (The NIH 2020 is available at the site: https://www.covid19treatmentguidelines.nih.gov/critical-care (accessed on 8 May 2020)). However, the Surviving Sepsis Campaign guidelines on the management of critically ill adults with Coronavirus Disease 2019 (COVID-19) recommended the use of empiric antimicrobials in mechanically ventilated patients with COVID-19 [24]. In Canada and Taiwan, empiric antimicrobials are administered within one hour in co-infection cases during severe acute respiratory infection and sepsis. This was also the case in the UK, where antibiotics were offered to suspected or confirmed COVID-19 cases with bacterial co-infection. Thus, in summary of the treatment outcomes, while some were successful and improved outcomes, others were not quite as helpful, raising questions on a common consensus about empiric treatment strategies. In the current study, we aim to identify the types, frequency, and antibiograms of SARS-CoV-2 co-infecting bacterial pathogens, as well as the clinical profiles, case aggravation, and/or correlation of co-infections with COVID-19 fatality rates (CFR) among tested patients.

## 2. Materials and Method

### 2.1. Study Designs

This was a retrospective cross-sectional study conducted at the King Salman Specialist Hospital (KSSH), Ha’il, Kingdom of Saudi Arabia (KSA). The hospital is a 500-bed tertiary care hospital designated to COVID-19 patients in addition to other specialized medical care services to Ha’il and all socioeconomic populations of the region. The hospital is certified and accredited by the Saudi Central Board for Accreditation of Healthcare Institutions (CBAHI) (Ref. No. HAL/MOH/HO5/34213). The KSSH has been designated to receive COVID-19 patients.

A panel of experts consisting of clinical microbiologists, laboratory specialists, and clinicians reviewed the list of patients for overall co-infection rates, COVID-19-compatible clinical profiles, and test results. COVID-19 patients (*n* = 301) were used to estimate the overall frequency and to monitor the rates of bacterial co-infections throughout the pandemic. We aimed to avoid co-infection bias, making sure that patients were not infected with different emerging variants, disease patterns, and isolate properties, as well as ensuring that the isolation period was the same for non-co-infected patients and co-infected patients. We then randomly selected 129 patients with bacterial co-infections. These included those currently discharged alive or who had died during hospitalization. The average ICU length of stay was ~17 days, even though it varied at different times. However, all COVID-19 cases at all times in KSSH were considered. All duplicate isolates and co-infections with similar or identical patterns were removed from the study. COVID-19 diagnosis for all participating patients was confirmed by clinically compatible symptoms and by real-time reverse transcription PCR (RT-PCR) testing performed on nasopharyngeal throat swab specimens at the Ha’il Health Regional Laboratory (HHRL) for COVID-19. The HHRL is a standard laboratory center, certified and accredited by the Saudi Central Board for Accreditation of Healthcare Institutions (CBAHI), Code 2739.

### 2.2. Data Collection Procedures

Electronic health records and medical microbiology laboratory reports of COVID-19 patients included in the study were retrospectively collected during the mid-quarter of 2021. Collection of data from KSSH, the COVID-19-designated hospital, included types of supportive oxygen therapies, demographics, laboratory results, lymphocyte counts, specimen types, antimicrobial susceptibility, and patient outcomes. Specific protocols used for all patients admitted with COVID-19 in this hospital are as follows.

1. Chest X-ray (CXR) was used for the detection of pulmonary involvement and monitoring the rapid progression of lung abnormalities in COVID-19, especially in critical patients admitted to the ICU following standard procedures. The CXR was performed either by a portable device or by using the departmental devices according to Jacobi et al. [25] and Borghesi and Maroldi [26]. Results of CXR were used along with PCR tests to designate test results as COVID-19-compatible.

2. Oxygen: Non-invasive ventilation procedures, such as supplemental oxygen, were used with patients with signs of severe respiratory distress, or hypoxemia (i.e., SpO2 < 90%). Initial oxygen therapy was used at 5 L/min and titrated to SpO2 ≥ 90%. If no improvement was seen, high oxygen flows (10–15 or 50–60 L/min) were delivered through a face mask with a reservoir bag to reach a higher concentration of oxygen according to Borghes and Maroldi [26] Nava et al., 2011 [27] and Keenan et al., 2011 [28]. Unless critical, the routine case management and oxygenation intervention strategies usually progress from simple to aggressive, as follows: First, a nasal canula (~4 L), then a simple facemask (~10 L), followed by a non-Rebreather mask (~15 L). If necessary, these are then followed by noninvasive medical ventilations for high flow, such as high-flow nasal canula (100 L) or Bilevel positive airway pressure (BiPAP). Often, tracheal intubation for high oxygen is required.

3. Intubation: Mechanical ventilations were required in patients who continued to have increased trouble with breathing or hypoxemia after using non-invasive ventilation. Invasive mechanical ventilation through an endotracheal tube or tracheostomy was performed by an ICU expert according to the NIH NHLBI ARDS Clinical Network’s mechanical ventilation protocol card, available at: http://www.ardsnet.org/system/files/Ventilator%20Protocol%20Card.pd (accessed on 5 December 2021).

4. Lowest absolute lymphocyte count (LALC): Routine complete blood and differential counts were performed by using the laboratory automated hematology analyzers according to Fan et al., (2020) [29] and Kaushansky et al., 2015 [30].

### 2.3. Microbial Co-Infection or Superinfection and Antimicrobial Susceptibility Data

Routine microbiological investigations for clinical co-infecting pathogens in different types of specimens, including sputum, urine, blood culture, culture of the respiratory tract secretions, swabs, and others, were conducted at the medical microbiology laboratory using standard bacteriology. Clinical co-infecting pathogens were identified by using routine standard bacteriological methods and susceptibility testing using automated systems. This included primarily the BD Phoenix system (BD Biosciences, Franklin Lakes, NJ, USA), MicroScan plus (Beckman Coulter, Brea, CA, USA), and the BD BACTEC system (BD Biosciences) for the identification and antimicrobial sensitivity analysis of microorganisms. Susceptibility was confirmed by culture and agar diffusion experiments. The susceptibility testing and breakpoint interpretive standards were carried out in accordance with the recommendations of the Clinical and Laboratory Standard Institute (CLSI document M100S-26 [31]. Test results obtained during the hospital stay were retrospectively collected and reviewed. Since patients with SARS-CoV-2 pneumonia sometimes lack respiratory secretions (~30% of them have sputum production, particularly in old age), specimen collection procedures were mostly performed in patients under invasive mechanical ventilation or with respiratory tract secretions. Co-infection during SARS-CoV-2 was considered when at least one of the performed microbiological investigations isolated a bacterial pathogen. Readjusting and de-escalation of the recommended Surviving Sepsis Campaign guidelines on empiric antibiotic therapy and the management of critically ill adults with COVID-19 followed as soon as microbiology results were available.

### 2.4. Statistical Analysis of the Data

Collected data were analyzed using Statistical Package for Social Sciences software (IBM SPSS; Version 24 SPSS version 23.0 for Windows, SPSS, Inc., Chicago, IL, USA). Descriptive and stratified analyses were conducted, and we present absolute numbers, proportions, and graphical distributions. We conducted exact statistical tests for proportions and show *p*-values where appropriate (a *p*-value < 0.05 was considered statistically significant).

### 2.5. Standard Definitions for Acquired Resistance as Multi-, Extremely, and Pan-Drug-Resistant Types (MDR, XDR, PDR)

Resistance classifications of MDR, XDR, and PDR were classified according to the guidelines of the European Centre for Disease Control [20]. The MDR was defined as acquired non-susceptibility to at least one agent in three or more antimicrobial categories, XDR was defined as non-susceptibility to at least one agent in all but two or fewer antimicrobial categories (i.e., bacterial isolates remain susceptible to only one or two categories), and PDR was defined as non-susceptibility to all agents in all antimicrobial categories [32].

### 2.6. Ethical Clearances and Institutional Review Board (IRB) Approval

Strict guidelines were followed during this research according to the Institutional Review Board (IRB) protocols. The ethical application for this study has been reviewed and approved by the Research Ethics Committee (REC at the University of Ha’il, KSA, dated 18 August 2020 and 22 October 2020, endorsed by University President letter number 55456/5/41, dated 29 December 1441 H, and 13/675/5/42 for projects RG191293 REC# H-2020-119 and RG20064 REC#H-2020-187. The King Abdulaziz City for Science and Technology (KACST)’s IRB registration numbers: H-8-L-074 IRB log 2020-29; 2021-11). However, most of the work was conducted on records of bacterial isolates, and all patients’ identities were removed from the study.

## 3. Results

A total of 129 patients were included in this study. The overwhelming majority of patients (71%) were seniors ≥50% years of age, while 25.6% were 21–49 years old and only 3% were 0–20 years old. The types of bacterial species, co-infection frequencies, and antibiograms patterns, as well as specimen sources and patients’ demographics, are shown in Table 1 (see details in Appendix A). All patients showed infiltration CXR findings. However, 35% required intubation (*n* = 45), and majority of these had *K. pneumoniae* co-infections (52%, *n* = 25) and did not survive, except for 3. Similarly, 22% of all patients who were intubated (*n* = 10) had *A. baumannii* co-infection and died. Based on recent resistance classifications, the antibiogram patterns of *A. baumannii* and *P. aeruginosa* were found to be defined as extremely drug-resistant nosocomial pathogens, while the enteric bacteria *K. pneumoniae* and *E. coli* were defined as multidrug-resistant (Table 1). The antimicrobials tested according to automated systems are shown in respective figures below. There is a good policy in place and not all of them were prescribed.

### 3.1. Antimicrobial Resistance Profiles in COVID-19 Co-Infections

*Klebsiella pneumoniae* was the most frequent bacterial nosocomial species found in this study, making up 37% of all bacterial species isolated. It primarily caused blood, urinary tract, respiratory tract, and other surgical wound infections. In addition, based on the standard resistance definitions, *K. pneumoniae* was classified as multi-resistant (Table 1). As shown in Figure 1, for 18 antimicrobials in different categories, over 50% of *K*. *pneumoniae* isolates were resistant, in some cases reaching higher resistances, such as AUG—amoxicillin*/clavulanic acid (2/1) (69.2%), ATM—aztreonam (67.3%), FOX—cefoxitin (66%), CRO—ceftriaxone (67.9%), AMC—ampicillin*/sulbactam (94%), CXM—cefuroxime (73.6%), KF—cephalothin (81.1%), and NIT—nitrofurantoin (64%). However, treatment options were still available with over 80% effectiveness in some others. A fair number of antimicrobials showed intermediate resistance, including tigecycline (TGC), NIT, tazobactam (TZP), imipenem (IMI), ciprofloxacin (CIP), and AUG.

*Acinetobacter baumannii* was the second most frequent nosocomial pathogen identified in this study. It was responsible for 26% of overall bacterial infections that were primarily isolated from blood and respiratory tract infections. *A*. *baumannii* was classified as extremely drug-resistant based on standard resistance classifications (Table 1). It was almost fully resistant to antibiotic treatment, except for colistin which showed effectiveness. Six antimicrobials showed full resistance to 100% of isolates, and these were: AUG100%—amoxicillin*/clavulanic acid (2/1), AMC100%—ampicillin, ATM100%—aztreonam, CXM100%—cefuroxime, FOX100%—cefoxitin, and ETP100%—ertapenem. However, for the rest of the antibiotics, *A. baumannii* isolates were nearly or over 90% resistant (Figure 2).

As shown in Table 1, *E. coli* was responsible for 18.6% of the overall infections which were primarily from the urinary tract. However, most of the patients were older and most were in the ICU. The antimicrobial susceptibility of *E. coli* was promising, and for most drugs used, a higher number of isolates showed susceptibility. The following antibiotics were 100% effective on isolates: namely, AK, CS, ETP, IMI, MRP, and TGC. Most of the resistances were lower than 50%, except for a few, such as AMC and KF, which were 69% and 66.6%, respectively (Figure 3). However, based on the recent standard definitions for the classification of drug resistance, *E. coli* was still found as multidrug-resistant in nosocomial infections.

*Pseudomonas aeruginosa* caused 8.5% of overall infections in this study. These were mostly from sputum specimens and swabs of respiratory tract and wound infections collected in the ICU and COVID-19 wards (Table 1). The antimicrobial susceptibility of *P. aeruginosa* showed a balanced pattern. Albeit it was classified as XDR based on recent drug classification guidelines, an almost equal number of drugs were found highly effective against this organism in in vitro assays. For instance, full resistance to the following antibiotics was observed: AUG, FOX, CRO, CXM, ETP, and TGC. However, higher levels of susceptibilities were observed in cephalothin (80%), ampicillin (86.6%), cefepime (85.7%), ceftazidime (80%), colistin (93.3%), and 73.3% each for gentamicin and imipenem (Figure 4).

Isolates of “Other” bacterial species were from different types of specimens, such as urine, blood, sputum, wound, and pleural effusions, etc., and they were categorized under the name “Other species” (Table 1, Figure 5). These species were less in number and did not show a specific pattern of disease. In addition, except for a few, they were highly susceptible to a wide range of antimicrobials used. These drugs included AK, FEP, CAZ, ETP, CIP, LEV, MRP, CN, TZP, and SXT. However, “Other” organisms showed higher resistances, reaching 100% resistance in some cases (e.g., AUG, AMC, CXM, KF, CS, and NIT). In addition, higher ranges of intermediate resistances were observed by isolates of the species in this group over all other species reported.

### 3.2. Clinical Profiles and Outcomes of COVID-19 Patient with Bacterial Co-Infections

The overall frequency of co-infection in the 301 consecutive cases of COVID-19 patients was 36% (*n* = 109). However, the CFR in patients with bacterial co-infection was 31.2%, compared to 9.9% in patients without bacterial co-infection (z-value: 3.1, *p*-value < 0.0001) (Figure 6). There was a significant association between bacterial co-infection and COVID-19 CFR. In other words, the survival rate of patients without co-infection was 90.1%, while with co-infection, 68.8% survived. In the 129 patients selected for focused co-infection patterns, 26% did not survive (*n* = 34).

Multidrug-resistant *K. pneumoniae* co-infection with SARS-CoV-2 was the most frequent (37%, *n* = 48) and showed strong correlations with death outcomes in COVID-19 patients (Table 1). All patients required ICU, were typically diagnosed with COVID-19 with PCR, as well as had COVID-19-compatible symptoms such as CXR and higher oxygen requirements through intubation. Among *K. pneumoniae* co-infected patients, 29% died (*n* = 14) (however data for nine patients’ outcomes were not available either for reasons of community infection and/or patient transfer). Major oxygen requirements were recorded irrespective of other intervention machines: 52% intubations (*n* = 25), 31% (*n* = 15) received >4 L oxygen, 33% ventilations (*n* = 16), of whom 88% died (*n* = 14), and 21% (*n* = 10) breathed normally. However, 14 patients had initial ventilation, then intubation and high oxygen, and all died. All patients had low LALC, mostly below 5 (see details in Appendix A).

Extremely drug-resistant *A. baumannii* was the second most frequent (26%, *n* = 34) co-infection for COVID-19 patients (Table 1) and was primarily isolated from blood and respiratory tract infections. All patients were in the ICU with COVID-19-compatible symptoms and CXR. Oxygen-supportive interventions were recorded in each machine, irrespective of other machines: three groups of patients, ten patients in each were either intubated, ventilated, or required >4 L, while nine breathed normally. However, 11 patients had initial ventilation and then were given high oxygen (>4 L) intubation, and all died later (33%, *n* = 11). All patients had low LALC, mostly below 5 (Table 1).

Multidrug-resistant *E. coli* co-infected 18.6% (*n* = 24) of COVID-19 cases, where majority were urinary tract infections in the ICU. Patients had typical COVID-19 symptoms, diagnostic CXR, and low levels of LALC (<5). For respiratory oxygen requirements, only three cases were ventilated, who were later intubated and then died. Patients’ oxygen requirements were >4 L. *Pseudomonas aeruginosa* was 8.5% of the overall isolates studied, mainly from wound and respiratory infections (Table 1 and see details in Appendix A). Based on standard definitions, it was classified as XDR; however, an almost equal number of drugs in tested categories were found highly effective against this organism. All patients were in the ICU or COVID wards with COVID-19-compatible CXR, and only one case of death had occurred among the two cases that were ventilated, and later intubated). Oxygen requirements for this species were higher than all other cases of bacterial co-infections (mostly 6–8 L). For other Gram-negative bacteria, five cases of deaths occurred, two each for Serratia and Morganella and one case of *Citrobacter* co-infection had occurred, who were all ventilated and later intubated.

## 4. Discussion

There has been a knowledge gap about the mechanisms of SARS-CoV-2 co-infection by bacterial pathogens during COVID-19 pandemic [33,34]. Determination of co-infection profiles is critical in deciding on the initial effective empirical therapy and patient management strategies. The role of co-infections in SARS-1 and MERS outbreaks were well-characterized, unlike that of SARS-CoV-2.

We found that bacterial co-infection with COVID-19 overall CFR (31.2%) was highly statistically significant in comparison to that of non-co-infected patients (9.9%). This is consistent with a recent finding in the Asir region in Southern Saudi Arabia, where higher mortality rates were associated with patients in the SARS-CoV-2 co-infection group compared to non-co-infected ones (50% vs. 18.7%, respectively) [35]. This also confirmed the earlier predictions that bacterial infections, particularly with highly resistant strains, are importantly associated with the COVID-19 outcome [36].

Here, we reported on an overall co-infection rate (36%, *n* = 109) among 301 COVID-19 patients. This is consistently lower than the widely reported rates during previous influenza outbreaks; however, it is similar to many SARS-CoV-2 co-infection reports, but lower than the rates reported in China and elsewhere [15]. In a UK secondary care setting, 27 (3.2%) of 836 patients had early bacterial co-infections in 0–5 days, which rose to 6.1% [37]. However, in Iran, 100% of COVID-19 patients were co-infected with *A. baumannii* (90%) and *S. aureus* (10%) [38]. Thus, global co-infection rates initially reported were quite variable, but mostly were low, reaching only as high as 50% among non-survivors, as reviewed by Lai et al. [3]. In contrast to many Gram-positive and -negative species reported in different countries, we found only a limited number (4 species: *K. penumoniae*, *A. baumannii*, *E. coli*, and *P. aeruginosa*, and a few various species). Of particular importance is that the former two species were significantly associated with aggravation and death rates. The limitation in Gram-positive bacterial populations in this study can be attributed to the rigid in-house MRSA screening upon and during admission, separate quarantine zones for COVID-19 and MRSA patients, and vigorous routine blood testing practices that are all in place since the last MRSA pandemic over a decade ago. In addition, our primary isolation sites were from acute respiratory secretions in COVID-19 zones compared to largely bloodstream isolates in [35]. Thus, the major differences in the sources of specimens may have resulted in different bacterial populations, albeit both studies are consistent in COVID-19 aggravation by mainly Gram-negative bacterial infections.

High prevalence of *K. pneumoniae* and *A. baumannii* and increased COVID-19 mortality contrasted with the multispecies co-infection reported by Ruan et al. [15]. Similar to the current study, *K. pneumoniae* has been found as a major species even among Gram-positives in other countries such as India (41.4% (79/191) [39]). In our study, the death rates in COVID-19 co-infected patients with the aforementioned pathogens, *Klebsiella pneumoniae* and *A. baumannii*, were 29% (*n* = 14) and 32% (*n* = 11), respectively. However, other enteric bacteria as well as the environmental pathogen *P. aeruginosa* co-infections did not correlate with increased fatalities, despite the latter being a major contributor to morbidity in cystic fibrosis and lung infections [40]. Additionally, patients who were co-infected with *K. pneumoniae* required aggressive oxygen interventions. For instance, 52% were intubated (*n* = 25), 31% (*n* = 15) received >4–5 L oxygen, and in those who required only supportive ventilation (33%), 81% died (*n* = 13). This potentially indicated selective aggravation by these two major species since survival rates were much higher (90.1%) in non-co-infected patients. However, it is not clear what the mechanism(s) of the selective prevalence in ventilated COVID-19 patients were. These findings support the emerging notion from different continents in which elderly males in the ICU are more prone to higher risks for co-infection with resistant infection, predisposing to mortality rates up to 50% [36,41,42,43,44]. However, acquired resistance alone, although it may aid in transmission and adaptation, would not account for the magnitude of the virulence per se. *Klebsiella pneumoniae* dominated despite its in vitro susceptibility to many antimicrobials prescribed. Evasion of drugs by intracellular dormancy and/or patients’ clinical resistance was not likely for the acute superinfection, despite specific treatment. Findings on selective co-infection were reported in prior influenza virus outbreaks, where pneumococcus experimentally impaired macrophage, neutrophil, and B cell responses to the virus [37]. In addition, at present, we do not have the evidence to suggest a *pneumoniae* hypervirulent strain. which have been singled out as an ‘urgent threat to human health’ by the virtue of sophisticated immune evasion strategies [45]. The global success of *K. pneumoniae* is usually attributed to its multidrug-resistance and hypervirulent pathotype; however, hypervirulence mechanisms are not clear in COVID-19 backgrounds. The World Health Organization (WHO) recommends ampicillin and gentamicin for the treatment of sepsis; however, increased prescription with a lack of specific data on infection, susceptibility, and patients’ responses to support these recommendations would only make matters worse worldwide [45,46,47,48]. Thus, selective COVID-19 co-infection of *K. pneumoniae* as well as the reductions in Gram-positive bacteria warrant further studies. Although strict patient screening protocols have been in place for *S. aureus* lineages, MRSA, CA-MRSA, VRS, etc., we believe additional factor(s) may have led to a specific outgrowth of selected Gram-negatives over others.

*Acinetobacter baumannii*, defined as extremely drug-resistant, was the second most frequent co-infecting pathogen (26%, *n* = 34); however, the mortality rate was slightly higher (at 33%, *n* = 11) than *K. pneumoniae*. All patients showed aggravated COVID-19-compatible symptoms in blood, respiratory tract, and CXR. There has been a spread of resistant Gram-negatives in the Arabian Peninsula since 2010 [49]. In a recent report, high COVID-19 co-infection rates by a carbapenemase-resistant strain of *A. baumannii* have been reported as causing outbreaks in a tertiary care referral hospital [50]. Furthermore, a recent report in Iran revealed a high prevalence of secondary carbapenem-resistant Gram-negative bacilli in COVID-19 patients admitted to two ICUs, revealing a high proportion of *K*. *pneumoniae* followed by *A. baumannii* during the first wave of the pandemic [51]. Thus, there is enough evidence to suggest a selective co-infection in hospitalized COVID-19 patients by limited Gram-negatives. Therefore, future molecular characterization is important for more insights into the profiles of these isolates. For oxygen requirements, 26.5% (*n* = 9) of patients were ventilated and then later intubated before they passed away, and some required >4 L, while another 26.5% breathed normally. Clinical intervention identified three equal groups (ten patients each) for the three procedures of intubation, ventilation, and those who received ≥4 L of oxygen, and another fourth group included patients who breathed without intervention. This was a less aggressive intervention compared to that of *K. pneumoniae*. Although *A. baumannii* has been listed by the WHO as a priority-1 pathogen due to enhanced virulence and resistance, much less is known about the virulence mechanisms, and it is spreading worldwide in COVID-19 patients, including in Wuhan (China), France, Spain, Iran, Egypt, New York (USA), Italy, and Brazil [52,53].

*E. coli* co-infected 18.6% (*n* = 24) of COVID-19 patients, had much better outcomes, and responded well to antibiotic options available for treatment, consistent with in vitro tests. *E. coli* prevalence rates were significantly different in geographic locations; for instance, it was predominant only in severely ill patients receiving treatment with invasive catheters in some countries [54,55]. However, in the list of the five most hospital-acquired bacterial superinfections diagnosed in Spain, *E. coli* was isolated from seven patients, ranking second to last, before *K. pneumoniae* [16]. Furthermore, despite its widely known virulence in lung infections, XDR *P. aeruginosa* was the least co-infecting pathogen (8.5%) in this study and was mainly isolated from wound and respiratory infections. In this study, only two cases were ventilated who were later intubated, and one of them subsequently died. However, *P. aeruginosa* was the most common pathogen responsible for ventilator-associated pneumonia (*n* = 26, 34.7%) [56]. Despite having the highest oxygen intervention (mostly 6–8 L), risk of patients’ advanced age, and its ability to adapt to lung infection and evasion of immunity [57], *P. aeruginosa* was associated with low death rates in this study. For “other bacterial species”, five cases of death, two each for Serratia and Morganella and one case of Citrobacter co-infection, occurred, who were all ventilated and later intubated. In addition, while most co-infections were typical, atypical bacterial species such as *Mycoplasma pneumoniae*, *Chlamydia pneumoniae*, and *Legionella pneumophila* have also been identified in different countries, including India [58]. However, while fungal infections were common in India, no evidence for concomitant fungal infections were identified in the UK study [37].

## 5. Conclusions

Taken together, we reported on the types, frequency, and antibiogram patterns of SARS-CoV-2 bacterial co-infections in hospitalized COVID-19 patients. We also studied clinical profiles and CFR in patients with and without bacterial co-infections. Four bacterial species were most common in COVID-19 patients, namely, *K. pneumoniae*, *A. baumannii*, *P. aeruginosa*, and *E. coli.* The first two species had the highest frequency of co-infection with SARS-CoV-2 in COVID-19 patients, which correlated with increased mortality in this study. We further found correlations of aggravated clinical profiles and CFR in patients with *K. pneumoniae* and *A. baumanni*. These findings have significant clinical implications in the successful empirical therapies of COVID-19 patients. Co-infections in COVID-19 patients may aggravate disease outcomes. Therefore, a better understanding of the types, frequency, and antimicrobial resistance profiles of co-infecting respiratory pathogens in COVID-19 patients can contribute to effective patient management and antibiotic stewardship during the current pandemic. Few laboratory data are available about the natural history of COVID-19 patients from the Middle East. The data from the KSA shed light on the evolving and expanding pandemic, where symptomatic patients tended to be older with more lymphopenia and worse outcomes [59], consistent with our findings. Some limitations of this study are worth mentioning. Since these are hospital-based data, a few cases may have been missed in the community if not admitted to the hospital, transferred to different units/clinics, or for non-consent issues. In addition, this was a single-center study of bacterial species with co-infection information from one Laboratory. A multi-center cohort study may provide more insights into the co-infection patterns and all types of microbial species.

## Figures and Tables

**Figure 1 ijerph-19-05270-f001:**
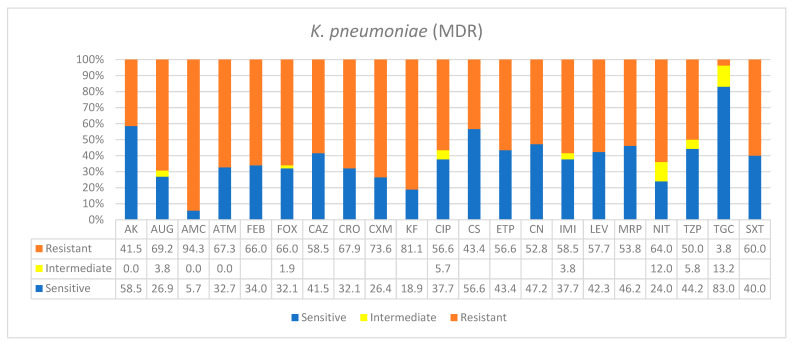
Antimicrobial sensitivity patterns of *K. pneumoniae* clinical isolates against 21 antibiotics. Abbreviations (in the order in which they appear in the figure): AK, amikacin; AUG, amoxicillin/clavulanic acid (2/1); AMC ampicillin*/sulbactam (2/1); ATM, aztreonam; FEP, cefepime; FOX, cefoxitin; CAZ, ceftazidime; CRO, ceftriaxone; CXM, cefuroxime; KF, cephalothin; CIP, ciprofloxacin; CS, colistin; ETP, ertapenem; CN, gentamicin; IMI, imipenem; LEV, levofloxacin; MRP, meropenem; NIT, nitrofurantoin; TZP, tazobactam; TGC, tigecycline; SXT, trimethoprim/sulfamethoxazole.

**Figure 2 ijerph-19-05270-f002:**
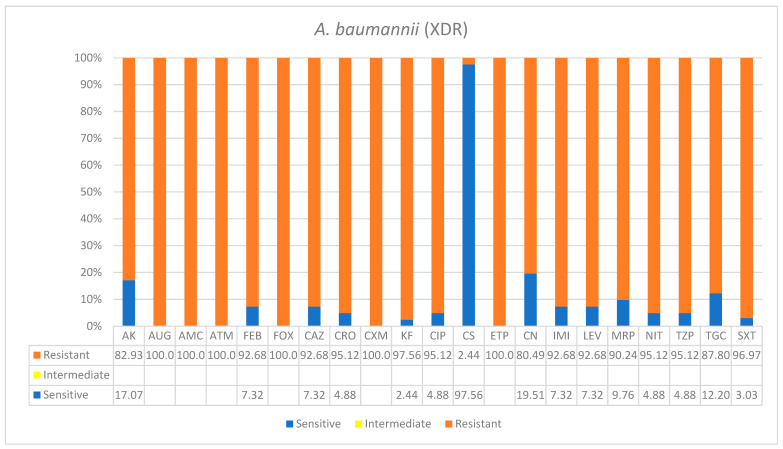
Antimicrobial sensitivity patterns of *A.baumanii* clinical isolates against 21 antibiotics. Abbreviations (in the order in which they appear in the figure): AK, amikacin; AUG, amoxicillin/clavulanic acid (2/1); AMC ampicillin; ATM, aztreonam; FEP, cefepime; FOX, cefoxitin; CAZ, ceftazidime; CRO, ceftriaxone CXM, cefuroxime; KF, cephalothin; CIP, ciprofloxacin; CS, colistin; ETP, ertapenem; CN, gentamicin; IMI, imipenem; LEV, levofloxacin; MRP, meropenem; NIT, nitrofurantoin; TZP, tazobactam; TGC, tigecycline; SXT, trimethoprim/sulfamethoxazole.

**Figure 3 ijerph-19-05270-f003:**
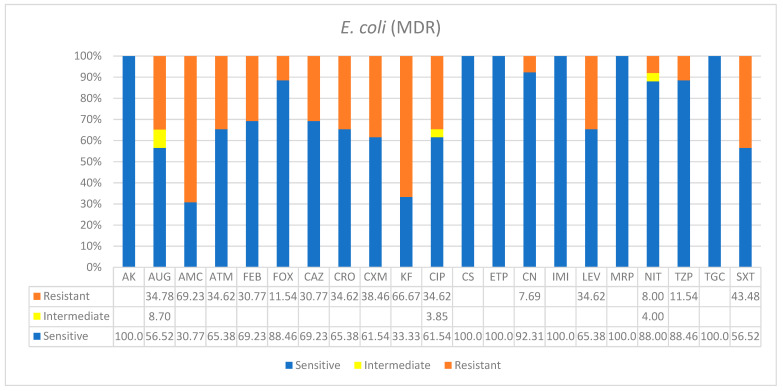
Antimicrobial sensitivity patterns of *E. coli* clinical isolates against 21 antibiotics. Abbreviations (in the order in which they appear in the figure): AK, amikacin; AUG, amoxicillin/clavulanic acid (2/1); AMC, ampicillin; ATM, aztreonam; FEP, cefepime; FOX, cefoxitin; CAZ, ceftazidime; CRO, ceftriaxone; CXM, cefuroxime; KF, cephalothin; CIP, ciprofloxacin; CS, colistin; ETP, ertapenem; CN, gentamicin; IMI, imipenem; LEV, levofloxacin; MRP, meropenem; NIT, nitrofurantoin; TZP, tazobactam; TGC, tigecycline; SXT, trimethoprim/sulfamethoxazole.

**Figure 4 ijerph-19-05270-f004:**
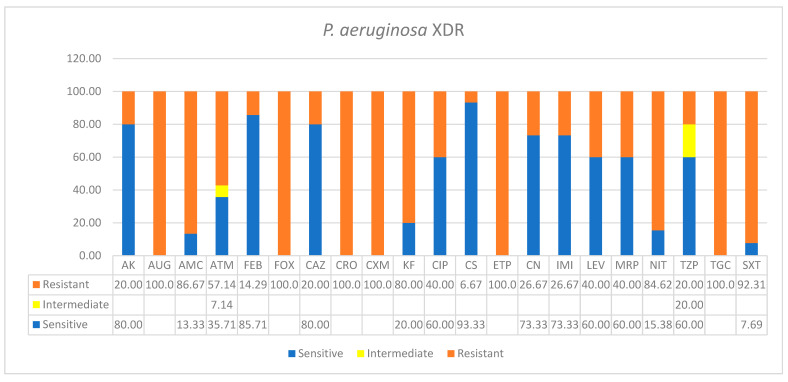
Antimicrobial sensitivity patterns of *P. aeruginosa* clinical isolates against 21 antibiotics. Abbreviations (in the order in which they appear in the figure): AK, amikacin; AUG, amoxicillin/clavulanic acid (2/1); AMC, ampicillin; ATM, aztreonam; FOX, cefoxitin; CAZ, ceftazidime; CRO, ceftriaxone; CXM, cefuroxime; KF, cephalothin; CIP, ciprofloxacin; CS, colistin; ETP, ertapenem; CN, gentamicin; IMI, imipenem; LEV, levofloxacin; MRP, meropenem; NIT, nitrofurantoin; TZP, tazobactam; TGC, tigecycline; SXT, trimethoprim/sulfamethoxazole.

**Figure 5 ijerph-19-05270-f005:**
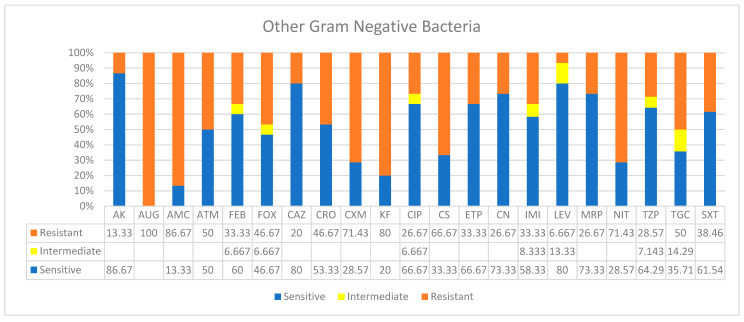
Antimicrobial sensitivity patterns of “other” Gram-negative bacterial clinical isolates against 21 antibiotics.

**Figure 6 ijerph-19-05270-f006:**
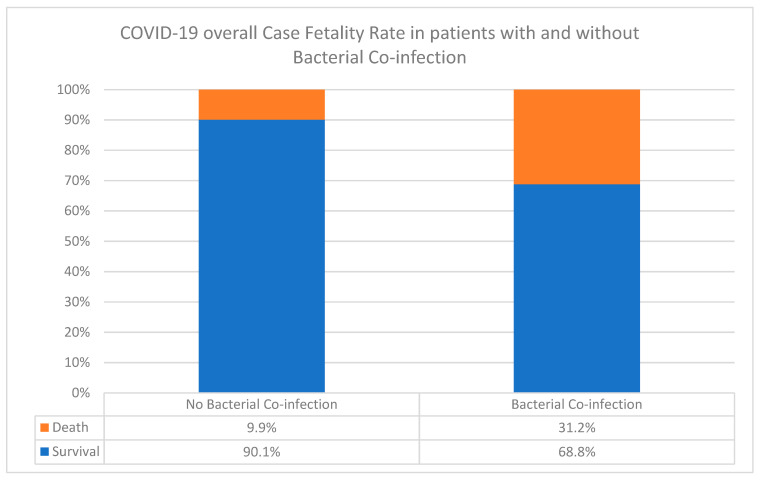
Overall COVID-19 case fatality rate in 301 patients with and without secondary bacterial co-infections in the Ha’il region, Saudi Arabia.

**Table 1 ijerph-19-05270-t001:** COVID-19 clinical profiles, bacterial co-infection, and patient demographics and outcomes in Ha’il, KSA.

Medical Ward	Bacterial Clinical Isolates (*n* = 129)									
	*K. pneumoniae* (*n* = 48)		*A. baumannii* (*n* = 34)		*E. coli* (*n* = 24)		*P. aeruginosa* (*n* = 11)		*Other Species* (*n* = 12)	
	*n*	%	*n*	%	*n*	%	*n*	%	*n*	%
ICU = Intensive Care Unit COVID = COVID-19 wards Overall isolates (*n* = 129)	48 Urine (11) Sputum (11) Blood (15) Wound/pus (8) Other (3)	37	34 sputum (13) blood (16), wound/pus (4)	26	24 Urine (16), Blood (3) Sputum (1) Wound (3) Screen(1)	18.6	11 Sputum (5) Swab/wound/pus (6)	8.5	12 Urine (4) Blood (1) Sputum (2) Wound (4) Pleural effusion (1)	9.3
MDR, XDR, PDR ^a^	MDR		XDR		MDR		XDR			
COVID-19 deaths in co-infected patients	29% (*n* = 14)		33% (*n* = 11)		12.5% (*n* = 3)		9% (*n* = 1)			
Number and types of oxygen support recorded for each specific bacterial co-infection (same patients often received different supports)										
# Intubations recorded	52% (*n* = 25) ^b^		29% (*n* = 10)		12.5% (*n* = 3) ^c^		18% (*n* = 2) ^d^			
Liters oxygen (>4 L)	31% (*n* = 15) ^b^		29% (*n* = 10)		12.5% (*n* = 3) ^c^		18% (*n* = 2) ^d^			
Ventilations recorded	*33*% (*n* = 16) ^b^		29% (*n* = 10)		12.5% (*n* = 3) ^c^		18% (*n* = 2) ^d^			
No breathing assistance	10 (21%)		26% (*n* = 9)		87.5% (*n* = 21)		82% (*n* = 9)			
LALC ^e^	<5		<5		<5		~6–8			
Age ^b^ in overall bacterial infections Ages affected in the four specific bacterial co-infections										
Young (1–20 years) overall 3% (*n* = 4)	6.25% (*n* = 3)		0		4% (*n* = 1)		0			
Adults (21–49 years) oveeall 25.6% (*n* = 33)	27% (*n* = 13)		26.5% (*n* = 9)		25% (*n* = 6)		18% (*n* = 2)			
Seniors (>50 years) 71% (*n* = 92)	64.6% (*n* = 31)		73.5% (*n* = 25)		71% (*n* = 17)		82% (*n* = 9)			
Overall Intubation rate	Yes, in 35% overall, most had *K. pneumoniae* and did not survive									
Overall Infiltration CXR	Yes, in almost all									

^a^ Standard classification as: MDR, multidrug-resistant; XDR, extremely drug-resistant; PDR, pan-drug-resistant. For *K. pneumoniae* (14 patients) and *A. baumanii* (11 patients), patients received all types of supports. ^b^ Age specific infections were same in both genders. ^c,d^ For *E. coli* and *P.aeroginosa*, the same patients received all supportive therapies. ^e^ LALC = Lowest absolute lymphocyte count.

## Data Availability

Not applicable.

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
