# Peer review of "COVID-19 Clinical Profiles and Fatality Rates in Hospitalized Patients Reveal Case Aggravation and Selective Co-Infection by Limited Gram-Negative Bacteria"

_ijerph, 2022, doi:10.3390/ijerph19095270_

Round 1
Reviewer 1 Report
It will be better if the authors could provide under normal condition(not CoVID) how about bacterial infection percentage or chance in hospitalized patients for a nice control.
Reviewer 2 Report
Dear authors, your manuscipt has a problem with English language and style. In addition, there are missing citations and the discussion can be improved. Methods need improvement too. I wish you best of luck but you must spend additional time on this draft. I software used by me detected "plagiarism" from this source: https://www.ncbi.nlm.nih.gov/pmc/articles/PMC8537776/ so please check and modify accordingly if needed.
Here attached the pdf with the editing.
Best,
the reviewer

Reviewer 3 Report
Please review my comments below;
- There is a repeat of this sentence “host macrophage hyperactivation” in line 95 and 96 of the introduction
- “Most of these (eight)….”, what does eight refer to “cases, patients” ?! please revise this sentence
- Please revise the writing of this manuscript to avoid contradictory information
- “correlated with increased death”, which correlation test did you use and was the correlation significant?
- The group report a variable rate of coinfection in CoV2patients than in different countries but did not discuss the possible underlying factors for this difference
Reviewer 4 Report
This manuscript by Kamaleldin B Said and co-workers reports a retrospective cross-sectional study carried out at the King Salman Specialist Hospital, Ha’il, (Kingdom of Saudi Arabia) about the impact of bacterial coinfections during COVID-19 disease. The types, the frequency, the antibiotic resistance patterns, the case fatality rates, and the clinical profiles of coinfecting microorganisms have been analysed in 301 COVID-19 inpatients. Bacterial coinfections have been observed in 109 (36%) SARS-CoV-2-infected patients and four drug-resistant Gram-negative bacterial species (Klebsiella pneumoniae, Acinetobacter baumannii, Escherichia coli, and Pseudomonas aeruginosa) were predominant. Case mortality rates were 31.2% in coinfected patients and 9.9% in non-coinfected patients, respectively. In addition, coinfections have mainly been observed in patients aged 50 years or older. The observed significant association between bacterial coinfection and mortality rates have important clinical implications in successful COVID-19 therapies; in particular, in the management of geriatric patients.
This article is similar to other co-infection studies carried out in different country but these analysis are however important in understanding how best to intervene to control bacterial superinfection in SARS-CoV-2-infected patients.
General comments:
The paper has some flaws.
Sometimes the text is difficult to follow. Are there 109 or 129 co-infected patients studied?
Why “20 cases of coinfections were added in the mid-quarter 2021 alone”?
Were different bacterial species present in the same patient?
The table that follows Figure 1 (please specify that it is Table 1 together with Table 2) shows 129 clinical isolates: 48 Klebsiella pneumoniae, 34 Acinetobacter baumannii, 24 Escherichia coli, 11 Pseudomonas aeruginosa, and 12 “other species”. It would be interesting to know which species.
Specific comments:
- Results
Figure 5
Sentence“Antimicrobial sensitivity patterns of “other Gram Positive” bacterial clinical isolates against 21 antibiotics”.
The sentence is imprecise because it is not "others" but exclusively Gram positives.
Furthermore, it would be interesting if the authors indicated the bacterial species they are describing.
Lines 366-368
“For other bacteria, five cases of deaths, two each for Serratia and Morganella and one case of Citrobacter coinfections had occurred who were all ventilated and later intubated.”
Serratia marcescens? Morganella morganii? Citrobacter spp.? Are gram-negative bacteria.
It is increasingly important that the authors indicate the gram positive bacterial species they have isolated.
Figure 6
Figure 6 also contains a table with the same results.
Please use either a Figure or a Table.
Supplementary materials: Please add “1” to the Table
References
A previous study on the effects of bacterial coinfections on intensive care unit (ICU)-admitted patients with COVID-19 in Asir province, Saudi Arabia also showed that gram-negative bacteria are the most commonly isolated bacteria in COVID-19 patients and that higher mortality rates are associated with patients in the coinfection group compared to the SARS-CoV-2- only infected group Alqahtani et al., 2022). This research needs to be pointed out, discussed, and the reference added.
Alqahtani A, Alamer E, Mir M, Alasmari A, Alshahrani MM, Asiri M, Ahmad I, Alhazmi A, Algaissi A. Bacterial Coinfections Increase Mortality of Severely Ill COVID-19 Patients in Saudi Arabia. Int J Environ Res Public Health. 2022 Feb 19;19(4):2424. doi: 10.3390/ijerph19042424. PMID: 35206609; PMCID: PMC8871991.
To improve the Discussion, other references need to be added.
For example:
- Bengoechea JA, Bamford CG. SARS-CoV-2, bacterial co-infections, and AMR: the deadly trio in COVID-19? EMBO Mol Med. 2020 Jul 7;12(7):e12560. doi: 10.15252/emmm.202012560. Epub 2020 Jun 15. PMID: 32453917; PMCID: PMC7283846.
- Silva DL, Lima CM, Magalhães VCR, Baltazar LM, Peres NTA, Caligiorne RB, Moura AS, Fereguetti T, Martins JC, Rabelo LF, Abrahão JS, Lyon AC, Johann S, Santos DA. Fungal and bacterial coinfections increase mortality of severely ill COVID-19 patients. J Hosp Infect. 2021 Jul;113:145-154. doi: 10.1016/j.jhin.2021.04.001. Epub 2021 Apr 20. PMID: 33852950; PMCID: PMC8056850.
- He S, Liu W, Jiang M, Huang P, Xiang Z, Deng D, et al. (2021) Clinical characteristics of COVID-19 patients with clinically diagnosed bacterial co-infection: A multi-center study. PLoS ONE 16(4): e0249668. https://doi.org/10.1371/journal.pone.0249668
- Fu, Y.; Yang, Q.; Xu, M.; Kong, H.; Chen, H.; Fu, Y.; Yao, Y.; Zhou, H.; Zhou, J. Secondary Bacterial Infections in Critical Ill Patients With Coronavirus Disease 2019. Open Forum Infect. Dis. 2020, 7, ofaa220.
- Baskaran, V.; Lawrence, H.; Lansbury, L.E.; Webb, K.; Safavi, S.; Zainuddin, N.I.; Huq, T.; Eggleston, C.; Ellis, J.; Thakker, C.; et al. Co-infection in critically ill patients with COVID-19: An observational cohort study from England. J. Med. Microbiol. 2021, 70, 001350
- Pourajam S, Kalantari E, Talebzadeh H, Mellali H, Sami R, Soltaninejad F, Amra B, Sajadi M, Alenaseri M, Kalantari F, Solgi H. Secondary Bacterial Infection and Clinical Characteristics in Patients With COVID-19 Admitted to Two Intensive Care Units of an Academic Hospital in Iran During the First Wave of the Pandemic. Front Cell Infect Microbiol. 2022 Feb 23;12:784130. doi: 10.3389/fcimb.2022.784130. PMID: 35281440; PMCID: PMC8904895. In Iran they found a high prevalence of carbapenem-resistant Gram-negative bacilli in COVID-19 patients admitted to our ICUs, with a high proportion of pneumoniae followed by A. baumannii.
- Elabbadi A, Turpin M, Gerotziafas GT, Teulier M, Voiriot G, Fartoukh M. Bacterial coinfection in critically ill COVID-19 patients with severe pneumonia. Infection. 2021 Jun;49(3):559-562. doi: 10.1007/s15010-020-01553-x. Epub 2021 Jan 3. PMID: 33393065; PMCID: PMC7779094. (They found a high prevalence of early bacterial coinfection during severe COVID-19 pneumonia, with a high proportion of aureus).
Round 2
Reviewer 2 Report
Dear authors, after the correction the paper is acceptable.
Best, the reviewer
Reviewer 3 Report
Non
Reviewer 4 Report
The authors replied satisfactorily to the referee's requests.